# Novel Cytoskeleton-Associated Proteins in *Trypanosoma brucei* Are Essential for Cell Morphogenesis and Cytokinesis

**DOI:** 10.3390/microorganisms9112234

**Published:** 2021-10-27

**Authors:** Marina Schock, Steffen Schmidt, Klaus Ersfeld

**Affiliations:** 1Molecular Parasitology, Department of Biology, University of Bayreuth, Universitätsstr. 30, 95447 Bayreuth, Germany; marina_schock@web.de; 2Computational Biochemistry, University of Bayreuth, Universitätsstr. 30, 95447 Bayreuth, Germany; steffen.schmidt@uni-bayreuth.de

**Keywords:** *Trypanosoma brucei*, cytoskeleton, microtubules, BioID, mass spectrometry

## Abstract

*Trypanosome brucei*, the causative agent of African sleeping sickness, harbours a highly ordered, subpellicular microtubule cytoskeleton that defines many aspects of morphology, motility and virulence. This array of microtubules is associated with a large number of proteins involved in its regulation. Employing proximity-dependent biotinylation assay (BioID) using the well characterised cytoskeleton-associated protein CAP5.5 as a probe, we identified CAP50 (Tb927.11.2610). This protein colocalises with the subpellicular cytoskeleton microtubules but not with the flagellum. Depletion by RNAi results in defects in cytokinesis, morphology and partial disorganisation of microtubule arrays. Published proteomics data indicate a possible association of CAP50 with two other, yet uncharacterised, cytoskeletal proteins, CAP52 (Tb927.6.5070) and CAP42 (Tb927.4.1300), which were therefore included in our analysis. We show that their depletion causes phenotypes similar to those described for CAP50 and that they are essential for cellular integrity.

## 1. Introduction

The cytoskeletal network of eukaryotic organisms is involved in almost all aspects of cellular processes [1]. However, the assignment of specific tasks has evolved very differently for individual components of the cytoskeleton both within organisms and between organisms. In metazoa, intermediate filaments, microtubules and filamentous actin are instrumental for shape, intracellular transport and locomotion, respectively. In protists, lacking intermediate filaments, microtubules are also often essential for cell shape and morphological integrity. This functional expansion, however, is unlikely due to, e.g., major differences in the way unit microtubules are built or assembled, as the sequences between tubulins of evolutionary highly diverse species and their in vitro properties are surprisingly conserved [2]. Key to these divergent functional trajectories is the emergence of specific sets of regulatory proteins [3,4,5,6]. An excellent model system to study such unique developments are kinetoplastids, and in particular the African trypanosome, *Trypanosoma brucei*, the causative agent of sleeping sickness.

The unicellular, flagellate parasite *Trypanosoma brucei* is a member of the excavates, one of the deepest branching groups of eukaryotes [7]. It is transmitted by the tsetse fly vector into the blood of the mammalian host. The process of changing between different hosts and different habitats within hosts requires complex morphological and metabolic adaptations [8].

The parasite’s microtubule-based cytoskeleton is crucial, not only for cell shape, but also for mitosis and organelle distribution during cell division [9]. The subpellicular cytoskeleton of trypanosomatids is composed of a nematic array of evenly spaced microtubules underneath the cell membrane. Several cytoskeleton-associated proteins (CAPs) link neighbouring microtubules to one another and to the plasma membrane [10,11]. However, although many CAPs have been identified in trypanosomes, their precise functions in the regulation and dynamics of the trypanosome cytoskeleton are largely unknown [4]. Some CAPs are also known to play an important role in cell division during which new microtubules are extended and intercalated between older microtubules [12,13]. One of these CAPs in *T. brucei* is the calpain-related protein CAP5.5, which is strictly regulated during the life cycle and only expressed in procyclic trypomastigotes [14]. The protein is N-terminally acylated with myristic and palmitic acid and cell biological and biochemical data support a model whereby CAP5.5 links the subpellicular microtubules, directly or indirectly, with the plasma membrane [14,15,16]. Depletion of CAP5.5, or its bloodstream-specific paralogue CAP8.1/CAP5.5V is lethal, causes aberrant cytokinesis, organelle mispositioning and partial disorganisation of the subpellicular microtubule array [15,16].

To identify proteins that are possible interactors of CAP5.5, the proximity-dependent biotin identification (BioID) technique was used [17]. This method uses a mutant version of the 35-kDa bacterial biotin ligase BirA* with low affinity for the highly reactive biotinoyl-5′-AMP intermediate produced in the active site of the enzyme. When BirA* is fused to a bait protein, the short-lived biotinoyl-5′-AMP labels amide groups of proteins in a proximity dependent manner. Even insoluble cytoskeletal proteins can be affinity purified and analysed as the biotin tag withstands harsh conditions for protein solubilisation [18].

In this work, BioID was used to identify possible binding partners and proteins localised in close proximity of CAP5.5. The protein found by this approach, named CAP50 based on its molecular mass, was characterised by analysing ectopic expression of a myc-tagged construct and RNAi-depletion. After detergent extraction CAP50 is, in vitro and in situ, associated with the trypanosomal cytoskeleton. The protein is essential for cell survival, and depletion resulted in a loss of structural cell integrity and cytokinesis defects. Circumstantial data that have emerged from several proteomics studies [19,20,21] suggest a possible functional link of CAP50 to two other, yet uncharacterised proteins, which we named CAP42 and CAP52. We show that these proteins are also associated with the subpellicular cytoskeleton. RNAi depletion results in similar phenotypes as observed after CAP50 depletion and they are also essential for cell division and maintenance of morphological integrity. Similar to almost all trypanosomal CAPs described so far, they have no similarities to microtubule-associated proteins in non-kinetoplastid organisms.

## 2. Materials and Methods

### 2.1. Trypanosome Cell Culture, Plasmid Construction and Transfection

In this study, the procyclic *T. brucei brucei* cell lines 449 (TetR BLE) [22] and 29-13 (T7RNA Pol NEO TetR HYG) [23] were used. The cells were grown in SDM-79 medium [24] supplemented with 10% *v*/*v* heat inactivated foetal bovine serum (Sigma-Aldrich, Taufkirchen, Germany), 5 µg/mL hemin (Sigma-Aldrich) and antibiotics (50 µg/mL hygromycin, 5 μg/mL bleomycin (InvivoGen, Toulouse, France) at 28 °C. 

Coding sequences were amplified using primers with incorporated restriction sites for cloning (for primer sequences, see Appendix A). To express CAP5.5 with a C-terminally fused BirA* in *T. brucei*, the BirA* module from pLEW100_myc_BirA (gift of Brooke Morriswood, University of Würzburg, Würzburg, Germany) [25] was first cloned into the pHD1800^2xmyc^ Vector (a derivative of pHD1700^2xmyc^ (gift of F. Voncken, University of Hull, Hull, UK) [26], multiple cloning sites modified with *Fse*I and *Asc*I recognition sequences) between the *Asc*I and *Pac*I sites. Subsequently, the CAP5.5 ORF was cloned between the *Fse*I and *Asc*I sites of the vector, resulting in a CAP5.5-myc-BirA*-myc construct. For ectopic protein expression open reading frames amplified from *T. brucei* genomic DNA were cloned into pHD1800^2xmyc^. RNAi was performed by cloning selected ORF fragments (https://dag.compbio.dundee.ac.uk/RNAit; accessed on 1 August 2021) of the proteins into the stem-loop vector pALC14 (gift of A. Schneider, University of Bern, Bern, Switzerland) [27].

For transfection, the vectors were linearised at their unique *Not*I site and stably integrated into the *T. brucei* rRNA spacer region by electroporation. A total of 3 × 10^7^ cells were resuspended in 100 µL transfection buffer (90 mM sodium phosphate, 5 mM KCl, 0.15 mM CaCl_2_, 50 mM HEPES, pH 7.3) and 10 µg vector DNA was added in 10 µL water. The suspension was transferred into an electroporation cuvette and transfected on an Amaxa Nucleofector II (program X-014, Lonza, Cologne, Germany) [28,29]. Cells were left to recover overnight in a flask with 10 mL SDM-79 medium and the antibiotics for selection were added the next morning (50 µg/mL hygromycin for pHD1800^2xmyc^ and 1 µg/mL puromycin for pALC14 transfected cells). Cells were cloned by limiting dilution in 96-well plates. Expression of transgenes was induced by addition of 1 µg/mL doxycycline to the medium. All restriction enzymes were from New England Biolabs, Frankfurt, Germany.

### 2.2. HeLa Cell Culture and Transfection

HeLa cells were maintained in high-glucose Dulbeccos’s Modified Eagle Medium (DMEM, Thermo Fisher Scientific) supplemented with 10% (*v*/*v*) heat-inactivated FBS and 1% (*v*/*v*) penicillin-streptomycin at 37 °C in a 5% CO_2_ humidified incubator. To transiently transfect HeLa cells the complete ORFs coding for the proteins were amplified from *T. brucei* genomic DNA using primers with *Fse*I and *Asc*I restrictions sites and cloned into the pCS2-FA-eGFP plasmid (gift of O. Stemmann, University of Bayreuth, Bayreuth, Germany). Cells were seeded onto coverslips in medium without antibiotics and allowed to attach overnight. For transfection medium was removed and 1.5 mL fresh DMEM without antibiotics added. In 150 µL OptiMEM medium (Thermo Fisher Scientific, Waltham, MA, USA) 3 µg plasmid DNA and 9 µg poly ethylenimine (Polysciences Inc., Warrington, FL, USA) were mixed by vortexing and incubated for 15 min at room temperature. This transfection mixture was carefully applied to the cells and incubated at 37 °C. After 6 h medium was removed, cells were washed with DMEM and allowed to grow overnight in DMEM without antibiotics. After a further 24 h cells were fixed with ice cold methanol for immunofluorescence.

### 2.3. Quantitative PCR

RNA was isolated using RNeasy plus mini kit (Qiagen, Hilden, Germany) according to the manufacturer’s instructions. Subsequently RNA was treated with DNAse to remove residual DNA by using the DNA-*free*™ DNA Removal Kit (Thermo Fisher Scientific,). cDNA was synthesised using the RevertAid First Strand cDNA synthesis kit (Thermo Fisher Scientific). For each 20 µL RT-reaction 500 ng RNA and the random hexamer primers were used. For qPCR cDNA was diluted to 1 ng/µL RNA transcribed and used in 20 µL reactions with SYBR GreenI (SYBR Green/ROX qPCR Mastermix, Thermo Fisher Scientific) and 0.6 µL gene specific primers (each 10 μM) (see Appendix A) in a Step One real time PCR system (Thermo Fisher Scientific). Primers were designed to amplify a gene-specific ~124 bp product and the cDNA of paraflagellar rod protein A mRNA was used for normalisation. RT reactions performed without reverse transcriptase and no-template reactions served as negative controls. qPCR was performed under the following cycling conditions: 95 °C 10 min, 40 cycles of 95 °C 15 s, 55 °C 30 s, 70 °C 30 s. Results were evaluated using the ∆∆Ct method with the Step One real time PCR software. For each qPCR experiment, data from two biological replicates, each consisting of three technical replicates, were averaged.

### 2.4. Fluorescence Microscopy

Trypanosomes were collected by centrifugation, pellets resuspended in PBS and settled on to poly L-lysine coated glass slides (VWR, Leuven, Belgium). Cytoskeletons were prepared from cells attached to poly L-lysine coated slides by extraction with 1% (*v*/*v*) NP-40 (Nonidet P-40) in PEM buffer (100 mM PIPES-NaOH, 1 mM MgSO_4_, 2 mM EGTA, pH 6.9). Whole cells and cytoskeletons were fixed with ice cold methanol for 20 min. Primary and secondary antibodies were applied for 1 h each in PBST (0.1% (*v*/*v*) Tween 20 in PBS). After incubations with antibodies, slides were washed three times for 5 min in PBST and finally incubated with 1 µg/mL DAPI in PBS for 5 min for DNA staining. Slides were mounted using Vectashield mounting medium (Vector Laboratories, Burlingame, CA, USA). Primary antibodies used: rabbit anti-GFP (Santa Cruz Biotechnology, Dallas, TX, USA); mouse anti-myc-tag (clone 9E10, DSHB, Iowa City, IA, USA); mouse anti-PFRA/C (L13D6; [30]); mouse anti-flagellum attachment zone (L3D2; [30]); rat anti-tyrosinated α-tubulin (YL1/2; [31]); mouse anti-trypanosomal-α-tubulin (TAT; [32]); mouse anti-α-tubulin (1:5000, kind gift of O. Stemmann). The antibodies L13D6, L3D6 and TAT were kind gifts of Keith Gull, University of Oxford, Oxford, United Kingdom. Secondary antibodies used: goat anti-mouse IgG-ATTO550 (Sigma-Aldrich); goat anti-mouse IgG-ATTO488 (Sigma-Aldrich); goat anti-rat IgG Alexa Fluor 488 (Thermo Fisher Scientific). Images were recorded on a Zeiss AxioImager M2 microscope (Zeiss, Oberkochen, Germany) equipped with a 100× PlanApochomat objective and a SPOT Pursuit monochrome CCD camera (Diagnostic Instruments, Burroughs, Sterling Heights, MI, USA). Image acquisition was done with VisiView software (Visitron, Puchheim, Germany). Pseudocolouring and overlays were done with ImageJ and figure assembly with Microsoft Powerpoint.

### 2.5. Protein Electrophoresis and Immunoblotting

For fractionation, 4 × 10^7^ PBS-washed cells were resuspended in 100 µL 1% (*v*/*v*) NP40 in PEM buffer (100 mM PIPES-NaOH, 1 mM MgSO_4_, 2 mM EGTA, pH 6.9) and incubated on ice for 2 min in the presence of EDTA-free mammalian protease inhibitor cocktail (Sigma-Aldrich). Lysates were centrifuged for 1 min at 20,000× *g*, supernatants were collected and precipitated with methanol and chloroform [33]. Pellets, containing the cytoskeletal fraction, were washed once in 1 mL PEM with 1% NP40. Pellets and dried precipitated supernatant fractions were dissolved in 50 µL hot 1× SDS-PAGE sample buffer (125 mM Tris-HCl pH 6.8, 5% (*v*/*v* glycerol, 4% (*v*/*v*) SDS, 5% β-mercaptoethanol, some crystals of Bromophenol Blue) and heated for 10 min at 99 °C. The equivalent of 4 × 10^6^ cells/lane was loaded, except for detection of tubulin by TAT antibodies where loading was 10^5^ cell equivalents/lane.

For immunodetection of proteins Western blots were blocked with blocking buffer (5% (*w*/*v*) skimmed milk in PBST) o/n. The primary antibody was diluted in blocking buffer and the secondary antibody in PBST with 1% (*w*/*v*) skimmed milk.

The following antibodies were used: mouse anti-myc-tag mAb (clone 9E10, DSHB); mouse anti-trypanosomal-α-tubulin mAb (TAT); mouse anti-BIP mAb (produced in our lab by G. Lallinger-Kube [34]); polyclonal mouse anti-CAP5.5 (1:500; produced in our lab by T. Demtröder and J. Jentzsch); rabbit anti-mouse-HRP (Sigma-Aldrich). 

Blots were blocked with 5% BSA/PBST for detection of biotinylated proteins and incubated with Precision Protein StrepTactin-HRP (Bio-Rad Labs, Feldkirchen, Germany) in 1% BSA/PBST for 1 h. Blots were developed with Ultra Sensitive HRP Substrate (Takara, Kusatsu, Japan) on an ImageQuant LAS-4000 (Fujifilm, Düsseldorf, Germany). 

### 2.6. Purification of Biotinylated Proteins and Mass Spectrometry

Cells were induced with 1 µg/mL doxycycline and incubated with 50 µM biotin for 24 h. After washing and extraction with 1% NP40, cytoskeletons were resuspended in lysis buffer (100 mM Tris-HCl, pH 8, 500 mM NaCl, 1 mM EDTA and 1× Protease Inhibitor Cocktail (EDTA-free) for use with mammalian cell and tissue extracts (Sigma-Aldrich)) and sonicated for 2 × 2 min on ice (small tip, 69% power, Bandelin Sonoplus, Berlin, Germany). The lysate was centrifuged (30 min at 22,000× *g*, 4 °C) and solubilised protein extracts were applied to a Strep-Tactin^®^XT Superflow^®^ gravity flow column (IBA GmbH, Göttingen, Germany). Biotinylated proteins were purified according to the manufacturer’s instruction using the Strep-Tactin^®^ Purification Buffer Set. 

For mass spectrometry bands of interest were cut from gels and trypsinised as previously described [35]. LC-ESI-MS/MS was performed on an LTQ-XL mass spectrometer (ThermoScientific) coupled with an EASY-nLC II chromatographic system (ThermoScientific) using an in-house packed column with ReproSil-Pur C18-AQ 3 µm beads (Dr. Maisch Gmbh, Ammerbuch, Germany). Protein identification was performed using Byonic software (Protein Metrics, Cupertino, CA, USA).

### 2.7. Transmission Electron Microscopy

RNAi cell lines were induced with 1 µg/mL doxycycline for 3 days and cells were pre-fixed in the growth medium with 2.5% glutaraldehyde (GA, EM-grade; Serva, Heidelberg, Germany) for 10 min at ambient temperature. Cells were washed once with Caco-buffer (0.1 M sodium cacodylate, 1 mM CaCl_2_, pH 7.2) and fixed with 2.5% GA/2% formaldehyde in Caco-buffer at ambient temperature for 60 min. After three 5-min washes in Caco-buffer cells were treated with 1% osmium tetroxide (Roth, Karlsruhe, Germany) in ddH_2_O for 120 min at 4 °C, followed by three 5-min washes in ddH_2_O. Cells were infiltrated with 20% BSA o/n at 4 °C, centrifuged for 10 min at full speed in a swing bucket microfuge, the supernatant carefully removed and the pellet fixed again with 2.5% GA in Caco-buffer for 4 h at 4 °C to obtain a firm pellet. The pellet was cut in small squares with a razor blade, washed in ddH_2_O and stained with 2% (*w*/*v*) uranyl acetate (Merck, Darmstadt, Germany) in ddH_2_O o/n at 4 °C. After three 5-min washes in ddH_2_O pieces were dehydrated in a graded ethanol series (30%, 50%, 70%, 95%, 3 × 100%) in 15–20 min intervals. After incubation in a 1:1 mixture of ethanol and propylene oxide and 2 × propylene oxide (Serva) for 15 min each, pieces were infiltrated with Epon: Epon/propylene oxide (1:3) 4 h at room temperature, Epon/propylene oxide (1:1) at 4 °C o/n, Epon/propylene oxide (3:1) 4 h at room temperature, 2× pure Epon (Serva) 4 h at room temperature and o/n. Each piece was transferred to a BEEM embedding capsule (Plano GmbH, Wetzlar, Germany), filled with fresh Epon and left to polymerise for about 2 days at 60 °C. Ultrathin sections (50–70 nm) were cut with a diamond knife (Diatom, Biel, Switzerland). Slices were mounted on copper grids (Plano GmbH) and post-stained with 2% (*w*/*v*) uranyl acetate for 30 min and lead citrate (80 mM sodium citrate (Fluka, Munich, Germany), 120 mM lead citrate (Merck) in 160 mM NaOH) for 3 min at room temperature in the dark and finally washed 4× with ddH_2_O. Sections were analyzed on a JEM-1400 Plus electron microscope (Jeol, Freising, Germany) equipped with a Ruby CCD camera (Jeol).

## 3. Results

### 3.1. Identification of Cytoskeleton-Associated Protein CAP50

To identify proteins directly or indirectly associated with CAP5.5 a proximity-dependent biotinylation assay (BioID) was used [25]. Procyclic PC449 cells were stably transfected with the pHD1800-CAP5.5-myc-BirA*-myc plasmid and the expression of the transgene after doxycycline induction was confirmed by Western blotting using an anti-myc antibody (Figure 1A). Anti-myc immunofluorescence microscopy of detergent-extracted cells displayed an even distribution of the fusion protein across the cytoskeleton (Figure 1B) as described for wild-type CAP5.5, excluding the flagellum (Hertz-Fowler et al., 2001). Consequently, the C-terminal fusion of CAP5.5 with BirA* did not interfere with correct subcellular localisation and did not cause any observable functional defects. To detect and purify CAP5.5 interacting proteins or proteins in close proximity of CAP5.5, transgene expression was induced in the presence of an excess of biotin. Cells were then detergent-fractionated into the cytoskeleton-containing pellet fraction (P) and a soluble fraction (S). Western blotting with HRP-conjugated streptavidin revealed several bands: one band at approximately 180 kDa, indicative for autobiotinylated CAP5.5-BirA* fusion protein plus two bands of around 70 kDa in the pellet fraction. An additional weak band at 70 kDa was detected in the soluble fraction, but not characterised further, assuming that binding partners of the abundant protein CAP5.5 are roughly present in equimolar amounts (Figure 1A). The pellet fraction was solubilised by sonication and the biotinylated proteins were affinity-purified. The fractions were separated by SDS-PAGE and the area of 60–70 kDa was cut out and analysed by mass spectrometry (Figure 1C).

This analysis identified a protein of 50.9 kDa which we named CAP50 (TriTrypDB ID is Tb927.11.2610) after its predicted molecular weight (the term CAP51 is already used for another protein [12]). CAP50 has previously been identified in a screen for flagellar attachment zone proteins as a cytoskeleton-associated protein [21], a proteomic analysis of isolated flagella [20], and BioID screens for proteins associated with the flagellar attachment zone proteins FPRC [19] and TOEFAZ1 [36], but in no case was it further characterised. Interestingly, in all screens that identified CAP50 as a cytoskeletal component, two other as yet uncharacterised proteins were identified by mass spectrometry. This association prompted us to include these proteins in our analysis. We termed the proteins CAP52 (TriTrypDB Tb927.6.5070) and CAP42 (TriTrypDB Tb927.4.1300) based on their calculated molecular masses.

### 3.2. CAP50 and CAP52 Are Related to Other T. brucei Cytoskeletal Proteins

A sequence analysis of the protein sequence of CAP50 revealed a structured N-terminal region that is predicted to be α-helical. Interestingly, six short repeats of approximately 12 amino acids can be identified in the unstructured C-terminus of the protein. A *T. brucei* database search using the repeat sequence identified *T. brucei* AIR9-like protein as the closest homologue (Appendix A). While having no overall similarity to CAP50, this protein contains five similar repeats in its unstructured C-terminus. A characteristic feature of the repeats is their strong negative charge (pI of the repeat regions is 4.6 for CAP50 and 4.2 for TbAIR9). Both repeat regions are predicted to be natively unfolded. These repeats are absent in *Arabidopsis thaliana* AIR9, where the protein was originally identified [37]. In contrast to AIR9 of *A. thaliana* and the AIR9-like protein of *T. brucei*, CAP50 does not contain Ig-fold A9 domains, nor does it contain the microtubule-binding serine-rich N-terminus, characteristic of the AIR9 protein of *A. thaliana* [37] and the leucine-rich repeats found in AIR9 and AIR9-like protein of *A. thaliana* and *T. brucei*, respectively [38]. The TbAIR9-like protein is also associated with the subpellicular cytoskeleton, excludes the flagellum and RNAi depletion in procyclic cells cause phenotypes resembling those described in this study for CAP50 (see below), such as aberrant cytokinesis and organelle mispositioning [39].

Sequence searches with CAP52 of *T. brucei* reveal a significant similarity (E-value = 9 × 10^12^) with a functionally uncharacterised protein, Tb927.9.11540 (Appendix A). This protein was also originally identified in a BioID screen for flagellar attachment zone proteins [36]. However, it has been shown, by endogenous tagging of the gene, to be associated with the subpellicular cytoskeleton. It displays a strong staining at the posterior end of the cell, fading towards the anterior end [36]. This pattern has also been confirmed by the TrypTag- project [40].

CAP42 shows no significant similarities to known cytoskeletal proteins in *T. brucei* or other organisms. However, it shows similarities to amidinotransferases (see TriTryp entry for Tb927.4.1300, E-value = 6.4 × 10^12^ against Pfam amidinotransferase domain). These enzymes are involved in a wide range of biosynthetic pathways in eu- and prokaryotes, from creatine synthesis in humans to antibiotics in bacteria [41,42]. CAP42 is the only *T. brucei* entry in the TrypDB database (https://tritrypdb.org/tritrypdb/app, accessed on 1 September 2021) annotated as a putative amidinotransferase.

### 3.3. Novel CAPs Are Associated with the Subpellicular Microtubule Cytoskeleton

To analyse if CAP42, CAP50 and CAP52 are generic microtubule-binding proteins, GFP-tagged proteins were transiently expressed in HeLa cells. None of the three proteins co-localised with microtubules (Appendix A). Whereas CAP42 and CAP50 were diffusely distributed in the cytoplasm, CAP52 formed inclusion body-like aggregates.

Ectopic expression of C-terminally myc-tagged versions of these proteins in *T. bruce*i was performed to further analyse the cytoskeletal association. Immunofluorescence analysis of detergent-extracted cells with anti-myc antibodies, recognising tagged CAPs, showed an even distribution of all proteins across the cytoskeleton, excluding the flagellum (Figure 2A). This is very similar to the localisation of all three proteins in whole cells as analysed by the TrypTag project (www.tryptag.org; accessed on 1 September 2021). Western blotting of NP40-fractionated *T. brucei* confirmed the cytoskeletal association of the proteins (Figure 2B). CAP42^2xmyc^ was also found in both the soluble and cytoskeletal cell fraction, which could be due to high levels of ectopic expression. All three proteins remained cytoskeleton-associated even in the presence of increasing NaCl concentrations (Figure 2C), similar to the well-characterised cytoskeleton-associated protein CAP5.5 (Figure 2C), indicating a high-affinity cytoskeletal association.

### 3.4. Depletion of CAP42, 50 and 52 Leads to Defects in Morphology, Cytokinesis and Growth

Based on qPCR, all three proteins are expressed both in the procyclic and bloodstream stages of the *T. brucei* life cycle. Transcript levels for both stages were similar for CAP50 while CAP42 and CAP52 mRNA levels where twice as high in procyclic cells compared to bloodstream cells (Figure 3A). To explore the function of the three CAPs, the proteins were depleted in procyclic cells using RNAi. Depletion levels of all three mRNAs were more than 95%, as analysed by qPCR (Figure 3A). After 2–3 days of RNAi, many cells displayed morphological abnormalities and cytokinesis defects (Figure 3B and Figure 4). RNAi-mediated protein depletion also led to strongly reduced proliferation for all three cell lines compared to non-induced and wild type cells (Figure 3C).

CAP42- and CAP52-depleted cells showed a moderate increase in anuclear zoids (1K) (10% and 6%) and multinucleate (>2N) cells (10% and 7%) accompanied by a slight decrease in 1N1K cells (Figure 3B). A concomitant increase in 2N2K cells indicated a delayed or defective cytokinesis. Even though, at population level, cells depleted of CAP42 or CAP52 had only moderately aberrant numbers of nuclei and kinetoplasts, but individual cells often showed an extremely altered phenotype (Figure 4B,C) compared to wild-type cells (Figure 4A). CAP52-depleted cells often exhibited asymmetric and incomplete divisions resulting in cells with multiple nuclei. The multinucleate cells often had low numbers of kinetoplasts (Figure 4B), indicating progressive karyokinesis in the absence of cytokinesis and basal body/kinetoplast segregation. 

The phenotype of CAP42 RNAi cells also showed severe impairment of cytokinesis, often associated with a rounded cell phenotype (Figure 4B). In some cells, the nucleus and kinetoplasts were mispositioned during mitosis with one kinetoplast anterior of the dividing nucleus instead of being located posterior or, during late anaphase, between the dividing nucleus (Figure 4C, YL1/2 staining).

CAP50-depleted cells showed similar phenotypes with multinucleated cells, rounded cells and a high proportion of zoids (46%) and multinucleate (>2N) cells (34%) (Figure 3B and Figure 4D). Additionally, and in contrast to CAP42- and CAP52-depleted cells, these cells often contained large numbers of kinetoplasts/basal bodies and flagella.

Formation of the flagellar attachment zone (FAZ) was not impaired in any of the three cell lines (Figure 4B–D) but flagella were sometimes found detached from the cell body.

### 3.5. CAP50 Is Essential for Subpellicular Cytoskeleton Integrity

CAP50 was identified by CAP5.5-BioID, suggesting a close physical association with CAP5.5, a protein shown to be essential for architectural integrity of the cytoskeleton [16]. Furthermore, CAP50 depletion revealed similar morphological phenotypes to those observed after CAP5.5 RNAi. Therefore, we were interested in a characterisation of this phenotype at ultrastructural level. Transmission electron microscopy of CAP50-depleted cells was performed to analyse the fine structure of the subpellicular microtubule arrays. Transverse sections exhibited mostly evenly spaced microtubule arrays underneath the plasma membrane, indistinguishable from wild-type cells (Figure 5IV). However, in some areas of the subpellicular microtubule arrays abnormally organised microtubules were found (Figure 5AI–III). Here, some microtubules had lost their close association with the plasma membrane and apposed microtubules and were localised further inside the cytoplasm. Again, this is very similar to defects in microtubule organisation of cells after CAP5.5 depletion [16].

To dissect the molecular basis of cytoskeleton association of CAP50, truncations lacking one-third of either the N- or the C-terminus were generated and their distribution examined in detergent-extracted cell extracts and cells. The truncated protein was detected in the cytoskeletal pellet fraction in ∆C-term^2xmyc^ expressing cells and in the supernatant in ∆N-term^2xmyc^ expressing cells, indicating that the N-terminus is responsible for targeting CAP50 to the cytoskeleton. This finding was also confirmed by immunofluorescence microscopy of detergent-extracted cells expressing the deletion constructs (Figure 5B). The ∆C-term^2xmyc^ construct was evenly distributed across the whole cytoskeleton, indistinguishable from the distribution of the full-length protein. In contrast, cytoskeletons of ∆N-term^2xmyc^ expressing cells were almost devoid of immunofluorescence staining (Figure 5B). 

To analyse whether the cytoskeleton association of CAP5.5 and CAP50 are, directly or indirectly, interdependent, CAP5.5 localisation was determined in CAP50-depleted cells extracted with increasing NaCl concentrations (Figure 5C). We observed that CAP5.5 started to dissociate from the cytoskeleton fraction at 0.5 M NaCl concentration in CAP50-depleted cells whereas it remained quantitatively associated with the cytoskeletal fraction in wild-type cells and appeared in the soluble fraction only at NaCl concentrations (>1.5 M), a concentration that starts to induce microtubule depolymerisation.

## 4. Discussion

We employed proximity-dependent biotinylation (BioID) to identify proteins associated or in close proximity of the cytoskeletal protein CAP5.5 and identified an as yet uncharacterised cytoskeleton-associated protein, CAP50. It was first identified in a proteomics analysis of the protein composition of intact flagella [20]. Subsequently, it was found in screens for flagellar attachment zone (FAZ) proteins or proteins functionally or spatially associated with FAZ proteins [19,21,36] and in a BioID screen for binding partners of *T. brucei* polo-like kinase TbPLK [43]. However, in none of those studies was a further characterisation of CAP50 pursued. The screen by Zhou et al. (2015), employing differential two-dimensional gel electrophoresis comparing wild-type with FAZ2-depleted cells, detected two additional proteins, CAP42 and CAP52, that were associated only with the subpellicular cytoskeleton and not with other cellular structures, such as the flagellum or organelles. This exclusive association with the microtubule cortical cytoskeleton is a common denominator that link CAP50, CAP42, CAP52 and CAP5.5. All these proteins have been identified in several different screens, mostly proteomics analyses of cytoskeletal elements, either of complex structures, such as flagella, or interacting partners of specific cytoskeletal proteins. A possible explanation for their appearance in flagellar proteome datasets is that it is very difficult to purify protein complexes resulting from screens targeting cytoskeletal elements to near homogeneity due to the insolubility under non-denaturing buffer conditions. They are then readily identified by highly sensitive mass spectrometry. Alternatively, the identified proteins are indeed functionally and spatially related. We showed that depletion of each of the three CAPs aborts successful cytokinesis, leading to multinucleated cells. Some of the previous screens specifically searched for proteins involved in regulation of cytokinesis using bait proteins known to be important for correct cell division [19,36,43]. For example, the protein TOEFAZ1/CIF1 is a component of the flagellar attachment zone, and its localisation is restricted to the tip of the growing new flagellum [43,44]. In a cell cycle-dependent manner, this FAZ protein co-localises with the TbPLK. The FAZ is also linked to the subpellicular microtubules via a multiprotein filament network [45,46]. Therefore, it is possible that proteins associated with the subpellicular cytoskeleton are involved in anchoring the FAZ to cortical microtubules. However, to our knowledge no proteins have been identified so far that are exclusively localised to microtubules juxtaposed to the entire length of the FAZ [3,4].

The protein sequence of CAP50 contains no informative motifs. The N-terminal two-thirds of the protein are predicted to be non-globular and to contain coiled-coil domains, typical for structural protein. The CAP50 C-terminus contains six imperfect repeats with a unit length of twelve amino acids. A biochemical feature of this repeat region is its highly negative charge under physiological conditions (pI 4.6, overall pI 5.1). This region is predicted to be structurally disordered. Such natively unfolded regions may adapt a rigid structure upon binding to a ligand, e.g., another protein [47]. Using deletion constructs, we have shown that the repetitive region is dispensable for cytoskeletal binding and therefore might be involved in interaction with other proteins. We have also shown that CAP50, in a heterologous mammalian cell culture system, does not bind directly to microtubules. The repeat region of CAP50 is similar to five repeats at the C-terminus of the cytoskeletal protein TbAIR9-like with a unit length of twelve amino acids [39]. This region in TbAIR9 is also strongly negatively charged (pI 4.2, overall pI 4.6). In contrast to CAP50, TbAIR9 is, with 110 kDa, much larger and contains additional motifs, such as the so-called A9 domains, predicted to form an immunoglobulin-fold structure, and N-terminal leucine-rich repeats. These features are in common with its plant homologue in *Arabidopsis thaliana*, which has been shown to be involved in the definition of the division plane during cytokinesis [38]. The plant AIR9 lacks the C-terminal 12 amino acid repeats. Negatively charged repeats are found in several cytoskeleton-associated proteins in *T. brucei*, such as autoantigen I/6, MARPs 1/2, Gb4 and WCB [48,49,50,51]. The repeats of MARP1/2 have been shown to directly mediate microtubule-binding [52], but the functions of the repeats found in I/6, WCB and Gb4 are unknown.

The second protein characterised in this study, CAP52, contains no motifs or predicted structural features that provide clues to a specific function. Structural prediction indicates that the protein is dominated by coiled-coil helices, indicating, as for CAP50, a structural role. Cap52 has significant homology with another, yet uncharacterised cytoskeletal protein, Tb927.9.11540. It is also predicted to have a coiled-coil dominated structure. The similarity extends over almost the entire sequence, suggesting a gene duplication event. In contrast to CAP52, which is distributed evenly over the entire cell body, Tb927.9.11540 is concentrated at the posterior half of the cell, forming a gradient with only a faint signal at the anterior cell pole [36]. Genome wide RNAi analysis showed a strong reduction in fitness/growth after depletion of both proteins in procyclic and bloodstream cells [53].

The third protein included in our analysis is CAP42. In contrast to CAP50 and CAP52, it is predicted to be a globular protein and BLAST analysis showed a significant similarity to amidinotransferases. Protein structure modelling using I-TASSER produced models with a reasonable confidence using known amidinotransferases as templates (data not shown). One of the most common reactions carried out by these enzymes is the deamidation of arginine to produce citrulline and ammonia. In certain bacteria, this is the first step in the anaerobic fermentation of arginine; in animals, it is involved in the biosynthesis of creatine or, as a peptidyl-amidinotransferase, in citrullination, a post-translational protein modification of arginine residues [41,54,55]. Because we demonstrated that CAP42 is a cytoskeleton-associated protein we speculated whether microtubules might be the target of this putative amidinotransferase. Using mass spectrometric analysis of proteolytically fragmented *T. brucei* tubulin, we were, however, unable to identify any citrulline residues instead of arginine (data not shown). Trypanosomes grown in culture scavenge citrulline from the serum and the growth of bloodstream form *T. brucei* at very low serum concentration requires the supplementation with citrulline [56]. This observation also argues against a function of CAP42 in citrulline synthesis.

The cellular phenotypes after RNAi-mediated depletion of CAP42, 50 and 52 are comparable. In all cases depletion is characterised by loss of cell shape. Cells lose their typical trypomastigote appearance and become distorted, sometimes rounded. This is accompanied by cytokinesis defects and organelle missegregation, resulting in multinucleate and anucleate cells. Such phenotypes are commonly observed after the depletion of cytoskeleton-associated proteins in trypanosomes. Typical examples are CAP5.5/5.5V [16], WCB [49], PAVE1 [36], CAP51/51V [12] or AIR9-like [39]. Given that, if analysed by genetic methods, depletion of all known cytoskeleton-associated proteins of *T. brucei* produce highly disruptive phenotypes and are most likely essential for viability in natural hosts, it is perhaps surprising that there is not a certain degree of robustness, such as functional redundancy, built into the cytoskeletal architecture. It appears that these proteins each contribute essential building blocks to the system’s integrity.

CAPs 42, 50 and 52 are probably not generic microtubule-binding proteins, as they fail to co-localise to microtubules after heterologous expression in HeLa cells. The sequences of α- and β-tubulin are extremely conserved between humans and trypanosomes and hence it seems unlikely that major structural differences prevent binding to human microtubules. Currently, we can only hypothesise that additional proteins are necessary to mediate cytoskeleton-binding or that unique combinations of posttranslational modifications, either on CAPs or microtubules or both, regulate binding. As with all cytoskeleton-associated proteins identified in trypanosomes so far, there are no significant similarities with known microtubule-binding motifs of mammalian MAPs. A direct binding to microtubules, assayed using mammalian microtubules in vivo or in vitro, has been shown only for few cytoskeleton-associated proteins of trypanosomes, such as CAP15/17 [57] and MARP-1/2 [48,52].

In conclusion, CAP42, 50 and 52 add to the extensive and, most likely, still incomplete list of essential proteins that contribute to the functional organisation of the microtubule cytoskeleton of *T. brucei*. It remains a challenge to assign specific roles and interactions for any of the proteins in this complex and highly interdependent structural assembly.

## Figures and Tables

**Figure 1 microorganisms-09-02234-f001:**
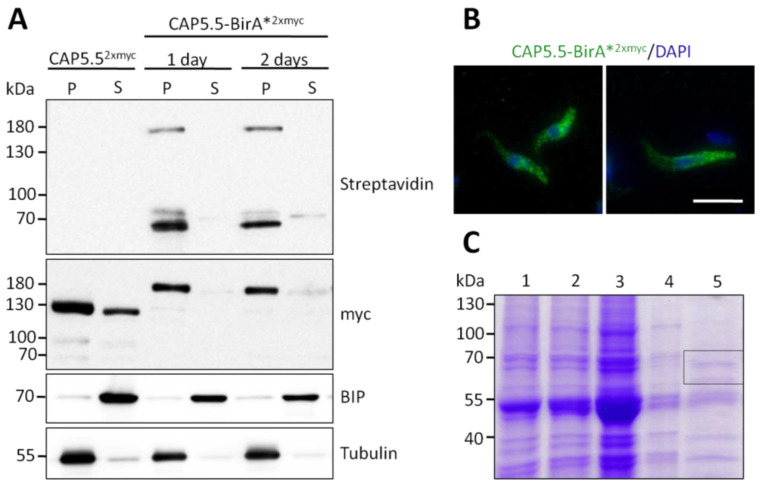
A CAP5.5-BirA*expressing cell line to identify CAP5.5 proximal proteins by BioID. (**A**) Ectopic expression of CAP5.5-myc-BirA*-myc (here simplified termed as CAP5.5-BirA*^2xmyc^) in procyclic trypanosomes (PC449) was induced by 1 µg/mL doxycycline with an excess of biotin in the culture medium. Samples were taken after 1 and 2 days and fractionated in the soluble (S) and the pellet (P) fraction using 1% NP40. Biotinylated proteins were detected by HRP conjugated streptavidin. CAP5.5-myc-BirA*-myc was detected using an anti-myc antibody. Cells ectopically expressing CAP5.5^2xmyc^ were used as control. BIP and tubulin served as the loading control for the soluble and the insoluble fractions. (**B**) Localisation of the CAP5.5-myc-BirA*-myc protein over the cell body in whole cells (left image) and 1% NP40 extracted cells (right image). Cells were stained using an antibody against the myc-tag and DAPI to stain the DNA. Scale bar, 10 µm. (**C**) Proteins of the pellet fraction were solubilised and biotinylated proteins were purified by a Strep-Tactin^®^ gravity flow column: 1, input, 2, flow through, 3, 4 wash fractions and 5, elution fractions pooled. The framed area of the gel, corresponding to the 60–70 kD mass range, was cut out and analysed by mass spectrometry.

**Figure 2 microorganisms-09-02234-f002:**
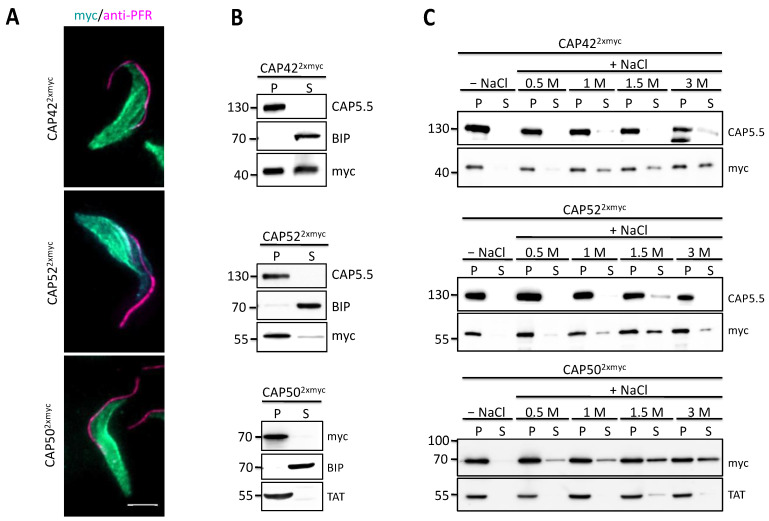
Characterisation of three cytoskeleton associated proteins. (**A**) Dox-inducible ectopic expression of C-terminally myc-tagged CAP42, CAP52 and CAP50 in *T. brucei*. The localisation of the CAPs to the cytoskeleton but excluding the flagellum was analysed by labelling 1% NP40 treated cells with anti-myc (green) and anti-PFR antibody (L13D6, magenta) to mark the flagellum. Scale bar = 5 µm (**B**) Dox-induced cells were fractionated with 1% NP40 and the soluble (S) and pellet (P) fractions were analysed for the myc-tagged proteins. Fractions corresponding to equal cell numbers were loaded in all lanes (**C**) The cytoskeleton-containing pellet fractions from C were treated with increasing amounts of NaCl and the supernatants (S) and pellets (P) were analysed for the myc-tagged proteins. Tubulin (TAT), CAP5.5 and BIP served as the loading control for the insoluble and soluble fraction, respectively. Representative blots of three biological replicate experiments are shown.

**Figure 3 microorganisms-09-02234-f003:**
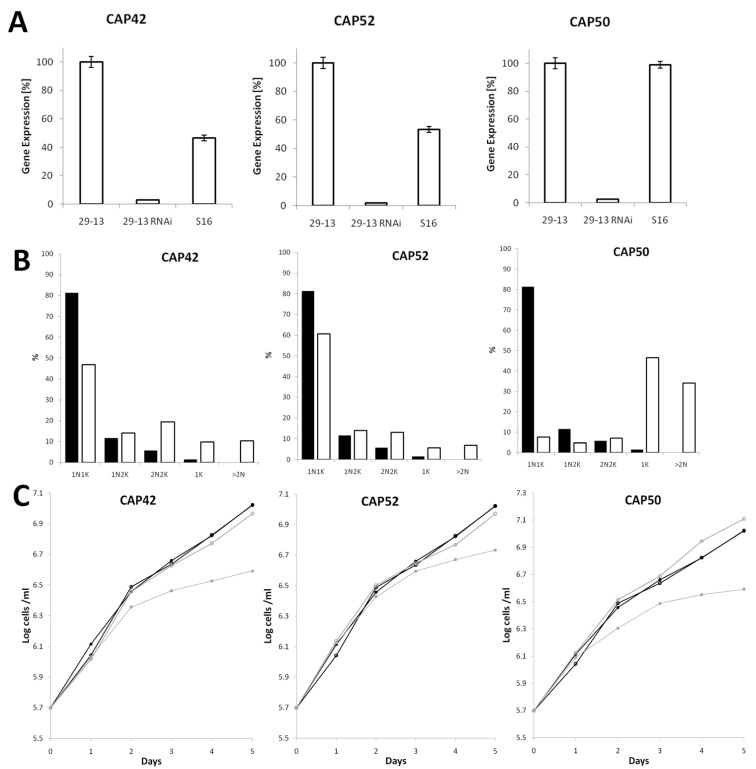
RNAi depletion of CAPs leads to reduced cell division and abnormal karyotypes in procyclic cells. (**A**) Relative mRNA expression levels of the three CAPs in procyclic (29-13), bloodstream (S16) and their respective RNAi cell lines (29-13 RNAi) measured by RT-qPCR. The RNAi cell lines were induced for 1 day. PFRA transcript was used for normalisation. (**B**) Analysis of cell cycle progression. Nuclei and kinetoplasts were visualised by DAPI staining. The number of nuclei (N) and kinetoplasts (K) was counted for each cell (*n* > 300). Black bars represent non-induced and white bars induced cells. Induction was for 3 days. (**C**) Cumulative cell growth of CAPs RNAi cell lines (grey lines, open circles: −tet; grey lines, closed circles: +tet) compared to wild type cells (black lines, open circles: −tet; black lines, closed circles: +tet). Growth curve data are averages of three independent clones.

**Figure 4 microorganisms-09-02234-f004:**
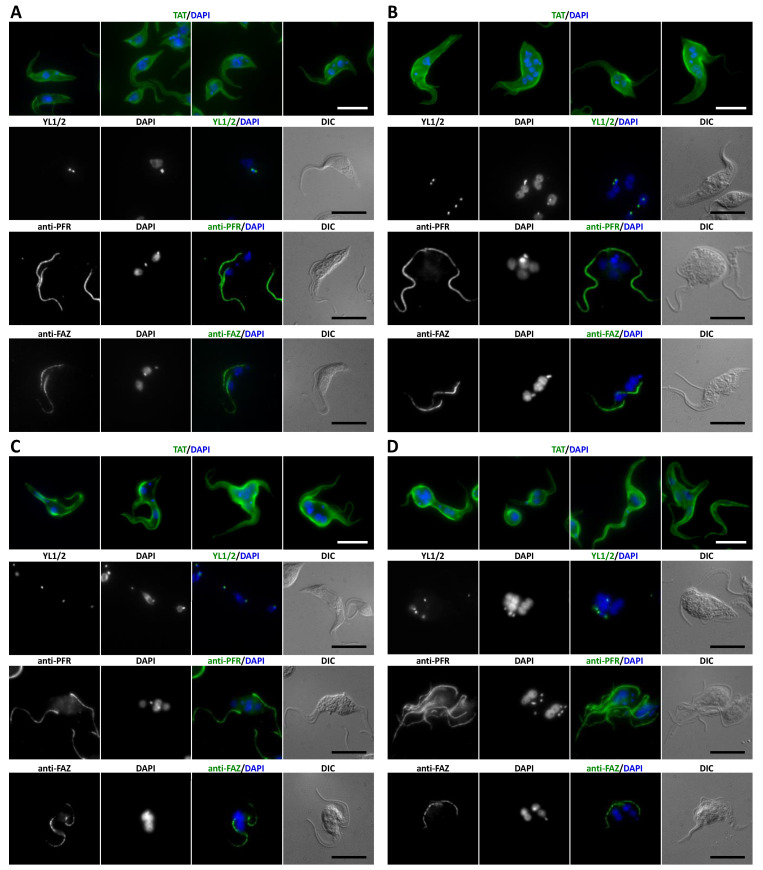
Immunofluorescence analysis of procyclic cells depleted of CAP52 (**B**) CAP42 (**C**) and CAP50 (**D**) by RNAi compared to wild-type 29-13 cells (**A**). Cells were doxycycline-induced for 3 days and fixed with methanol. TAT: anti-α-tubulin, YL1/2: anti-tyrosinated-α-tubulin (marker for basal bodies), PFR: anti-paraflagellar rod proteins A/C (L13D6), FAZ: anti-flagellum attachment zone (L3D2). DNA was stained with DAPI. Scale bar = 10 µm.

**Figure 5 microorganisms-09-02234-f005:**
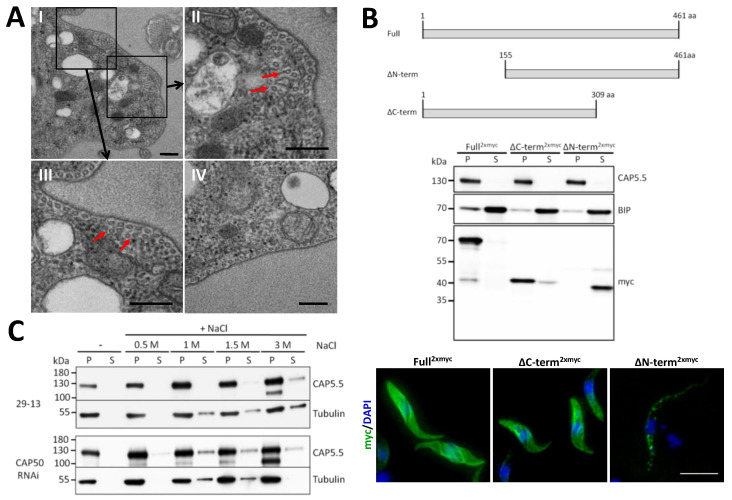
Ultrastructural and biochemical characterisation of CAP50. (**A**) TEM of transverse sections through procyclic cells depleted of CAP50 after RNAi induction for 3 days. Boxed areas in panel (I) are magnified in panels (II) and (III), showing examples of disordered microtubule arrays (some marked by red arrows). Panel (IV) shows a cross-section of a wild-type cell. Scale bar, 200 nm. (**B**) Truncation constructs of CAP50 used to analyse contribution of the C- or the N-terminus to cytoskeleton association of the protein. Truncation constructs and the full-length protein as control were C-terminally 2xmyc-tagged. For Western blot analysis, induced cells were fractionated with 1% NP40 in pellet (P) and supernatant (S) and the distribution of the protein was detected with an anti-myc antibody. Anti-BIP and anti-CAP5.5 were used as loading controls for the soluble and insoluble cytoskeleton fractions, respectively. For immunofluorescence cytoskeletons were labelled with an anti-myc antibody and DAPI. Scale bar, 10 µm (**C**) Analysis of the impact of CAP50 depletion on CAP5.5 cytoskeleton association. CAP50 RNAi cells were induced for 3 days and extracted with 1% NP40. The cytoskeletal pellet fraction was subjected to increasing concentrations of NaCl and supernatant (S) and pellet (P) fractions were analysed for their CAP5.5 content by Western blotting. Anti-tubulin was used to track the integrity of the microtubules. Wild type cells (29-13) were used as a control. Equal cell number equivalents were loaded in each lane. Representative blots of three biological replicate experiments are shown.

## Data Availability

Not applicable.

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
