# Peer review of "Novel Cytoskeleton-Associated Proteins in Trypanosoma brucei Are Essential for Cell Morphogenesis and Cytokinesis"

_microorganisms, 2021, doi:10.3390/microorganisms9112234_

Round 1

Reviewer 1 Report

Microorganisms-1421795-peer-review-v1

Article  Novel Cytoskeleton-Associated Proteins in Trypanosoma brucei Are Essential for Cell Morphogenesis and Cytokinesis. Marina Schock, Steffen Schmidt, and Klaus Ersfeld. 
In this manuscript, the authors show their work on the animal protist pathogen Trypanosoma brucei and have focused on identifying and characterizing novel sub-pellicular cytoskeletal associated proteins - aka CAPs. They used proximity biotinylation (BioID) as a test on a well-characterized trypanosome cytoskeleton-associated protein CAP5.5 where essentially, CAP5.5 was used as the “bait” to identify proteins that are in close proximity. Using this approach they identified CAP50 and based on bioinformatics analysis they correctly predicted the localization of two other cytoskeleton binding proteins, CAP52 and CAP42 to the cytoskeleton.  They show that depletion of any of these proteins by RNAi induces the production of phenotypes similar to those for CAP5.5 RNAi and demonstrate that all three proteins are essential for cellular integrity and cell survival.

In my opinion, this is a well thought out logical, and well-written manuscript with clear objectives, well-executed experiments, and comprehensible analysis of the results.

Subsequently, I do feel that with some minor experimentation this work is sufficient for publication in Microorganisms, but I would like the authors to address the following major and minor comments first;

Major

I only have one major point; Line 389, the authors wrote that; “In order to analyze whether the cytoskeleton association of CAP5.5 and CAP50 are interdependent, CAP5.5 localisation was determined in CAP50-depleted cells extracted with increasing NaCl concentrations (Figure 5C)”. 

I am not sure that this is the best way to test this hypothesis because high salt could be extracting unknown CAP proteins that may influence CAP5.5 or CAP50 binding to microtubules or other MAPs. Nevertheless, the opposite experiment, where CAP5.5 is knocked down and CAP50 is monitored, was not done and this should be done to confirm the hierarchy or dependency relationships of these two CAP proteins.

Minor

Line 295, the sentence is not clear; “For none of the three proteins co-localization with microtubules was found”.  Indeed it sounds odd. Please rewrite it.

Line 433 is grammatically confusing. It reads “We showed that depletion of all three CAPs aborts successful cytokinesis…”. This could mean all three simultaneously or each one independently? From what I understand only one CAP was depleted at any one time, not all three, so this sentence needs a rewrite.

Line 441. The authors state that; “However, to our knowledge, no proteins have been identified so far that are exclusively localized to microtubules juxtaposed to the FAZ, such as the antiparallel microtubule quartet [3, 4]. Actually, if I understand this sentence correctly then it is inaccurate because Spef1 (DOI: 10.1074/jbc.M112.417428 ) and proteins identified by BioID of Spef1 (DOI: https://doi.org/10.1128/mBio.00668-20), are located on the MTQ. Even though they do not run the full length of the MTQ they are localized on them. Please correct this.

The resolution of all the images is quite poor. Hopefully, this is not the publication level resolution.

Author Response

The reviewer is right to point out that testing the interdependence of binding of CAP50 and CAP5.5 salt extraction of proteins does not necessarily probe direct interactions but also indirect interactions, mediated by other proteins. Since we had only 5 days to return the experiment, we didn’t have the opportunity to address this question by further experiments. Instead, we have rephrased our interpretation to take this possibility into account.

All other comments of this reviewer addressed minor issues, which we have all rectified. One comment related to the data on Spef1, which was identified as a MTQ-associated protein. However, the data also show that this protein is probably associated with basal body structure and extend a only into the proximal region of the MTQ. It does not, and this was our point, associate exclusively and along the entire length with the MTQ. We have rephrased this to make this clear.

Concerning the image resolution, we have submitted separate image files with the first submission. I do not know whether they will be used in the final version. However, we have, in the resubmission, the images inserted into the text with new versions. They seem to have a sufficient resolution. In addition, we will again submit, if possible, all images as separate high resolution files.

Reviewer 2 Report

SUMMARY

This paper from Schock et al. describes the identification and initial characterisation of three novel components of the microtubule corset in Trypanosoma brucei. The authors initially carried out a proximity labelling screen using the microtubule corset-associated protein CAP5.5 as a probe, and identified an uncharacterised protein (which they named CAP50) as a hit. Based on CAP50 being identified alongside two other uncharacterised proteins in similar screens performed by other labs, they decided to investigate all three proteins (CAP50, CAP52, CAP42) here. They confirmed that all three proteins tightly associate with the cytoskeleton, and then probed the effects of their depletion using a battery of cell biology assays.

The manuscript is concise, detailed, and well-written, and does not require significant alteration. There are some interesting data here, and the story as a whole constitutes an interesting addition to the literature and will be fruitful to follow up in future work, I think.

I have made a few suggestions that I think might help improve the manuscript, but there is nothing of concern here, and the authors are free to exercise their discretion as to how many points they choose to implement.

MAJOR POINTS

- Reproducibility information needs providing for all figures. How many separate clones (biological replicates) were used, and how many independent experiments were carried out.

MINOR POINTS

- L153 Details of microscopy acquisition are essential. Please provide these (microscope, objective lens, acquisition software, processing/analysis software, and so on).

- Fig1. Was there any biotinylation of tubulin seen? It's not possible to assess this based on the cropped blot in Fig1A, but Fig1C suggests that tubulin was a hit. This might be worth mentioning as it's an obvious positive control when using proximity labelling for a microtubule corset protein.

- Conclusions on the effects of RNAi are based solely on the presentation of exemplary cells (Figure 4). These conclusions would be more compelling if some of the effects were quantified and the proportions either quoted in the text or presented as plots.

- The observation of bilaterally symmetric phenotypic cells (best examples in Fig4B anti-PFR panels, Fig 4C all panels, Fig 4D YL1/2 and anti-FAZ panels), with two fully replicated flagella and FAZ, is extremely interesting. It suggests that cell division proceeds normally, including the ingression of the cleavage furrow, up until the furrow reaches the end of the FAZ. The cells appear to have been unable to resolve the segregation of their posterior ends. Quantifying the incidence of such cells might be very informative, and consideration of what might produce them might be an interesting addition to the discussion.

EDITORIAL POINTS

Abstract - none.

Introduction

L28 - "protist" instead of "protozoa" is preferable nowadays. "Protozoa" relates to an outdated system of classification in which all protists were labelled as either primitive animals or primitive plants (protophyta).

L37 - the clade Excavata no longer exists in the latest eukaryote phylogeny (Adl et al., 2019). I would recommend instead writing "...brucei is a member of the excavates, one of the..."

L42 - parasite's not parasites

L49 - I would perhaps also recommend citing Sinclair et al., 2021 here ("The Trypanosoma brucei subpellicular...").

Materials and Methods

L82 - It's a bit odd that the citations just from 21 all the way to 44 in this section, without the intervening papers being cited in between.

L95 - Maybe mention how the ORF fragments were selected (RNAit?).

L102 - overnight not o/n.

L138 typo: "PBS and settled"

L140 - NP40 can be either Nonidet P-40 (aka IGEPAL CA-630) or Tergitol type NP-40. Please specify which was used.

L140 - The counterion used to set pH of the PIPES is not specified. This should be PIPES-NaOH, right?

L143 - Is the 0.1% Tween20 really w/v and not v/v?

L155 - see notes on L140.

L157 - was the inhibitor cocktail EDTA-free? This should be specified if so.

L166 - skimmed not skim.

L173 - Some details on blot visualisation (ECL?) and data recording (digital scanner?) would be helpful.

L176 - see notes on L140.

L177 - Again, specification of whether the protease inhibitor cocktail is EDTA-free would be really helpful here. Many cocktails contain 2 mM EDTA, which would change the final concentration of this chemical in the buffer.

L198 - swing bucket not swing out.

Figures/figure legends:

- All figures require the addition of information on reproducibility. How many separate clones (biological replicates) were used, and how many independent experiments were done.

Figure 1

Fig1B

- do the authors have a negative control (i.e image of the -dox condition)? This would increase confidence in the specificity of the labelling.

- the length of the scale bar is not disclosed. This could be added to the panel.

- the blue LUT has poor contrast against the black background - the authors might consider switching to using cyan instead.

Fig 1C

- it's fantastic that the authors have shown all the different fractions instead of just the elution, but it would be even more informative if they provided information on the % size of each. They are clearly not the same % fraction, as the signal from lane 3 (wash) is greater than lane 1 (input).

Figure 2

Fig2A

- Using red and green LUTs is not compatible with colour blindness, and in addition red has poor contrast against a black background. I would strongly recommend switching to magenta and green instead.

- The antibody label at the top of the panel is incorrect: L13D6 not D13L6.

- Presenting the exemplary cells in the same orientation would make it easier to compare the labelling patterns.

Fig2B

- Were equal fractions of S and P loaded? This should be noted if so.

Figure 3

Fig 3A

- How long post-induction was this analysis carried out?

- The inclusion of a single exemplary electrophoresis image would be helpful.

- The use of dot plots (e.g. with the PlotsOfData web app) would be a more information-rich visualisation than the bar charts used here.

Fig 3B

- This is one set of experiments where the disclosure of the number of independent experiments is really critical.

- How long post-induction was this analysis carried out?

- Would it be possible to calculate the values for each experiment and then present averages. At the moment there are no error bars and so it's not possible to tell how much each category varies.

Fig 3C

- Again, the disclosure of separate clones and independent experiments would strengthen this set of experiments.

- It is a bit hard understanding which line is which - instead of a description in the figure legend (grey/black lines, open/closed circles), could this information be shown graphically in an inset in one of the charts?

Fig 4

 - Typo: "FAZ: anti-flagellum" not "ant-flagellum"

- The I /II /III / IV notation needs explaining in the figure legend.

- The use instead of cyan instead of blue LUT for the DAPI signal would provide better contrast.

Fig 5

Fig 5A

- Labels I / II /III /IV are not really visible. Maybe they could be placed inside white boxes?

- The panel description mentions red arrows but I don't see any.

Fig 5B

- Describing these as truncation instead of deletion constructs might be more precise.

- Were equal fractions loaded? This should be mentioned if so.

- The length of the scale bar in the micrographs is not disclosed.

- Consider using cyan instead of blue LUT for DAPI.

Fig 5C

- Were equal fractions loaded?

Results

L223 - The citation style has changed here.

L247 - It would probably be worth mentioning what annotation the TrypTag project has assigned to all three proteins. This would be an additional level of confirmation.

L263 - Revealed, not reveals.

L328 - If the growth defect is described before the cell cycle counts, then shouldn't this be Fig 3B instead of Fig 3C?

L379 - As noted above, I think "truncations" is more precise than "deletions" here.

Brooke Morriswood

University of Würzburg

Author Response

We are very grateful to Brooke Morriswood for the detailed review of the manuscript and for pointing out many small omissions and ambiguities.

We have added reproducibility and statistical information to figures were appropriate.

Details of microscopy acquisition has been added

Tubulin was indeed visible on the blots, but only as a minor labelled component, indication a critical distance to the bait protein. Indeed, so far no one has shown that CAP5.5 binds directly to microtubules. Tubulin peptides were also major hits in mass spectrometry identification of purified proteins.

The reviewer has provided a long list of suggestions for minor, editorial changes. We have complied with most suggestions, such as changing the colour scheme of some images for better visibility, adding some more details into Materials and Methods (NP-40, protease inhibitors, imaging).

I should point out that the expression data in Fig. 1C are based on RT-qPCR. Therefore, no electrophoresis was involved and no gels are shown.

Concerning the growth curve lines in Fig. 3C, we had added the symbols into the legends, but after changing to the required font (Palatino Linotype) of the journal, this was technically no longer possibility and we had to use text. We would have preferred our original solution.

We have also added the information that, on western blots, of course equivalent cell numbers were loaded in each lane of the corresponding gels.

Concerning a reference to TrypTag, we were a bit cautious, because TryTag did not extract cells to obtain cytoskeletons. They used whole cells and therefore a localisation of a protein to the cell body was not that informative. However, all three proteins do localise to the cell body, excluding the flagellum, thus confirming our data. We have added this information to the text.